# Ring-like oligomers of Synaptotagmins and related C2 domain proteins

Maria N Zanetti[1], Oscar D Bello[1], Jing Wang[1], Jeff Coleman[1], Yiying Cai[1], Charles V Sindelar[2], James E Rothman[1]*, Shyam S Krishnakumar[1]*

[1]Department of Cell Biology, Yale School of Medicine, New Haven, United States; [2]Department of Molecular Biophysics and Biochemistry, Yale School of Medicine, New Haven, United States

**Abstract** We recently reported that the C2AB portion of Synaptotagmin 1 (Syt1) could self-assemble into $Ca^{2+}$-sensitive ring-like oligomers on membranes, which could potentially regulate neurotransmitter release. Here we report that analogous ring-like oligomers assemble from the C2AB domains of other Syt isoforms (Syt2, Syt7, Syt9) as well as related C2 domain containing protein, Doc2B and extended Synaptotagmins (E-Syts). Evidently, circular oligomerization is a general and conserved structural aspect of many C2 domain proteins, including Synaptotagmins. Further, using electron microscopy combined with targeted mutations, we show that under physiologically relevant conditions, both the Syt1 ring assembly and its rapid disruption by $Ca^{2+}$ involve the well-established functional surfaces on the C2B domain that are important for synaptic transmission. Our data suggests that ring formation may be triggered at an early step in synaptic vesicle docking and positions Syt1 to synchronize neurotransmitter release to $Ca^{2+}$ influx.

*For correspondence: james.rothman@yale.edu (JER); shyam.krishnakumar@yale.edu (SSK)

**Competing interests:** The authors declare that no competing interests exist.

## Introduction

Synchronized rapid release of neurotransmitters at the synapse is a highly orchestrated cellular process. This involves maintaining a pool of synaptic vesicles (SV) containing neurotransmitters docked at the pre-synaptic membrane, ready to fuse and release their contents upon the influx of calcium ions ($Ca^{2+}$) following an action potential, while also preventing the spontaneous fusion of SVs in absence of the appropriate cue (*Südhof and Rothman, 2009*; *Jahn and Fasshauer, 2012*; *Südhof, 2013*; *Rizo and Xu, 2015*). The core machinery required for the $Ca^{2+}$ triggered neurotransmitter release are the SNARE proteins (VAMP2, Syntaxin, and SNAP25) as well as Munc13, Munc18, Complexin and Synaptotagmin (*Südhof and Rothman, 2009*; *Jahn and Fasshauer, 2012*; *Südhof, 2013*; *Rizo and Xu, 2015*). A combination of biochemical, genetic and physiological results have pinpointed Synaptotagmin as a central component involved in every step of this coordinated process (*Wang et al., 2011*; *Jahn and Fasshauer, 2012*; *Südhof, 2013*; *Rizo and Xu, 2015*). The principal neuronal isoform, Synaptotagmin 1 (Syt1), is a SV-associated protein, with a cytosolic domain consisting of tandem $Ca^{2+}$-binding C2 domains (C2A and C2B) attached to the membrane via a juxtamembrane 'linker' domain (*Brose et al., 1992*; *Takamori et al., 2006*).

Accordingly, Syt1 acts as the immediate and principal $Ca^{2+}$ sensor that triggers the rapid and synchronous release of neurotransmitters following an action potential (*Brose et al., 1992*; *Geppert et al., 1994*; *Fernández-Chacón et al., 2001*). Upon $Ca^{2+}$ binding, the adjacent aliphatic surface loops on each of the C2 domains partially insert into the membrane and this enables the SNAREs to complete membrane fusion by mechanisms that are still uncertain (*Tucker et al., 2004*; *Rhee et al., 2005*; *Hui et al., 2006*; *Paddock et al., 2011*). Syt1 is also needed for the initial stage of close docking of SVs to the plasma membrane (PM), requiring in particular the interaction of the polybasic region on C2B domain with the anionic lipid, phosphatidylinositol 4, 5-bisphosphate (PIP2)

**eLife digest** Reliable communication between neurons is essential for the brain to work properly. This is accomplished by tightly controlling how chemical messengers, called neurotransmitters, move between neurons. Neurotransmitters are typically packaged into bubble-like structures called synaptic vesicles and are released only when the neuron receives an input electrical signal. A set of proteins orchestrates the release of the neurotransmitters from the neuron, which happens after the synaptic vesicles fuse with the cell membrane.

Synaptotagmin, a protein found on the surface of the synaptic vesicle, plays many roles in neurotransmitter release. It helps to attach the synaptic vesicle to the cell membrane and also prevents the vesicles from fusing to the membrane in the absence of an appropriate input signal. Most importantly, it detects when the electrical signal arrives at the neuron by binding to calcium ions that flood the cell following the input signal. This triggers the rapid fusion of the vesicles to the cell membrane. It is not clear how Synaptotagmin is able to carry out its different roles and in particular, control how neurotransmitters are released as calcium ions enter the cell.

Zanetti et al. have now used a technique called negative stain electron microscopy to investigate how Synaptotagmin molecules taken from mammals arrange themselves on the surface of a membrane. In this technique, individual Synaptotagmin proteins on the surface of a synthetic membrane are chemically marked and their structure is imaged using an electron beam. Using this approach under conditions resembling those in cells, Zanetti et al. found that 15–20 copies of Synaptotagmin came together and formed ring-like structures on the membrane surface. These ring structures were rapidly broken apart when calcium ions were added to them.

Further investigations suggest that the ring structures form when synaptic vesicles first attach to a membrane. Overall, it appears that the Synaptotagmin rings act as washers or spacers to prevent the vesicle from fusing to the cell membrane until the rings are disrupted by the arrival of calcium ions. Future studies are now needed to investigate whether the ring structures form inside cells and whether they act together with other proteins involved in neurotransmitter release.

at the PM (*Bai et al., 2004*; *Wang et al., 2011*; *Parisotto et al., 2012*; *Park et al., 2012*; *Honigmann et al., 2013*; *Lai et al., 2015*). The C2B domain also binds to the neuronal t-SNAREs (Syntaxin/ SNAP25) on the PM, which positions the Syt1 on the pre-fusion SNARE complexes and contributes to the docking of the SV but is by itself insufficient (*de Wit et al., 2009*; *Parisotto et al., 2012*; *Mohrmann et al., 2013*; *Kedar et al., 2015*; *Park et al., 2015*; *Zhou et al., 2015*).

Despite a wealth of information on Syt1 function and underlying molecular mechanism, critical questions remain. Deletion (or mutations) of Syt1 eliminates fast synchronous release and increases the normally small rate of asynchronous/spontaneous release (*Geppert et al., 1994*; *Littleton et al., 1994*; *Bacaj et al., 2013*). Reciprocally, removing Complexin increases the spontaneous release amount and the remaining Syt1 is only capable of mounting asynchronous release, though this release is still $Ca^{2+}$-dependent (*Huntwork and Littleton, 2007*; *Hobson et al., 2011*; *Jorquera et al., 2012*; *Cho et al., 2014*; *Trimbuch and Rosenmund, 2016*). This suggests that Syt1, acting in concert with Complexin, also functions as a clamp to both restrain and energize membrane fusion to permit rapid and synchronous release (*Giraudo et al., 2006*; *Krishnakumar et al., 2011*; *Kümmel et al., 2011*). How this clamping is accomplished still remains a mystery. In addition, fast neurotransmitter release exhibits a steep cooperative dependency on $Ca^{2+}$ concentration, which implies that several $Ca^{2+}$ ions need to be bound to one or more Syt1 molecules to trigger release (*Schneggenburger and Neher, 2000*, *2005*; *Matveev et al., 2011*). Further, reduced $Ca^{2+}$ binding affinity does not change this $Ca^{2+}$ cooperativity (*Striegel et al., 2012*), suggesting multiple copies of Syt1 molecules might be involved in gating release. However, the exact mechanism of the cooperative triggering of SV fusion is unclear.

We have recently shown that Syt1 C2AB domains can form $Ca^{2+}$-sensitive ring-like oligomers on phosphatidylcholine (PC)/phosphatidylserine (PS) lipid surfaces (*Wang et al., 2014*). This finding suggests a simple and elegant mechanism: If these Syt1 ring-like oligomers were to form at the interface between SVs and the plasma membrane, they could act sterically to prevent fusion, until this barrier

is removed when $Ca^{2+}$ enters and triggers ring disassembly i.e. the Syt1 ring would synchronize fusion to $Ca^{2+}$ influx. In addition, the oligomeric nature of Syt1 could explain the observed $Ca^{2+}$ cooperativity of neurotransmitter release. Here we show that the ring-like oligomer is a common structural feature of the C2 domain containing protein and describe the physiological correlates of the Syt1 ring oligomer which argues for a functional role for the Syt1 ring in orchestrating the synchronous neurotransmitter release.

## Results

### Circular oligomeric assembly is a common feature of C2 domain proteins

We had previously described the formation of $Ca^{2+}$-sensitive ring-like oligomers on lipid monolayers with the C2AB domain of Syt1 (*Wang et al., 2014*). To explore this further, we analyzed the organization of membrane bound C2AB domains of other neuronal isoforms of Synaptotagmin (Syt2, Syt7 and Syt9) on lipid surface under $Ca^{2+}$-free conditions by negative stain electron microscopy (EM). Syt2 and Syt9 act as $Ca^{2+}$ sensors for synchronous SV exocytosis but are expressed in only a subset of neurons (*Xu et al., 2007*), while Syt7 has been posited to mediate the $Ca^{2+}$-dependent asynchronous neurotransmitter release (*Bacaj et al., 2013*). EM analysis on lipid monolayer was carried out as described previously (*Wang et al., 2014*). Briefly, the lipid monolayer formed at the air/water interface was recovered on a carbon-coated EM grid and protein solution was added to the lipid monolayer under $Ca^{2+}$-free conditions (1 mM EDTA) and incubated for 1 min at 37°C. Negative-stain analysis revealed the presence of ring-like oligomers for all the Syt isoforms tested (*Figure 1*). Despite the variability in the number of ring-like structures between different isoforms, the size of the ring oligomers were remarkably similar, with an average outer diameter of ~30 nm (*Figure 1*). In all cases, each ring was composed of an outer protein band of a width of ~55Å, which is consistent with the dimensions of a single C2AB domain (*Fuson et al., 2007*). This data shows that the ability to form the circular oligomers is not unique to Syt1, but conserved among the Syt isoforms and further suggests that it might be an intrinsic property of the C2 domains.

Therefore, we next tested the C2AB domains of Doc2B, C2ABCDE domains of extended Synaptotagmin 1 (E-Syt1) and the C2ABC domains of E-Syt2. Doc2B is a C2 domain protein expressed in the pre-synaptic terminals and a putative $Ca^{2+}$ sensor that regulates both spontaneous (*Groffen et al., 2010*) and asynchronous release (*Yao et al., 2011*). E-Syts are endoplasmic reticulum (ER) resident proteins, which contain multiple C2 domains and have been implicated in ER-PM tethering, the formation of membrane contact sites, and in lipid transport and $Ca^{2+}$ signaling (*Giordano et al., 2013*; *Reinisch and De Camilli, 2016*; *Fernandez-Busnadiego, 2016*; *Herdman and Moss, 2016*). Doc2B and E-Syt2 formed circular oligomeric structures on lipid monolayers analogous to those seen with Syt isoforms (*Figure 1*). However, we observed very few and unstable ring-like oligomers with E-Syt1 (*Figure 1*). The lack of ring-like oligomers for E-Syt1 might be due to the insufficient concentration of this protein on the membrane surface as E-Syt1 has very weak affinity to the membrane under $Ca^{2+}$-free conditions (*Idevall-Hagren et al., 2015*).

The uniform dimensions of the ring oligomers of the multi-C2 domain proteins suggested that the ring is formed by a single C2 domain, with the other C2 domain(s) projecting away radially (*Figure 1*). This implies that the ring oligomerization is not a general property of all C2 domains, but only a select few. Consistent with this, we find that the Syt1$^{C2B}$ domain alone can form the ring-like oligomers albeit a bit smaller in size, but the Syt1$^{C2A}$ cannot (*Figure 1—figure supplement 1*). Brief treatment of the pre-formed ring oligomers with 1 mM $Ca^{2+}$ (*Figure 1—figure supplement 2*) revealed that all of the Syt isoforms (Syt1, Syt2, Syt7, and Syt9) and Doc2B were sensitive to $Ca^{2+}$ and are rapidly disrupted, but E-Syt were either un-affected (E-Syt2) or even stabilized (E-Syt1). Altogether, our data suggests that ring-like oligomers are a common structural feature of C2 domain containing proteins, but their sensitivity to $Ca^{2+}$ is divergent (discussed below in detail).

### Complete cytoplasmic domain of Syt1 forms rings under physiologically relevant conditions

To assess the functional relevance of the Syt1 ring oligomers, we sought to understand the molecular aspects of the oligomer assembly and the $Ca^{2+}$ susceptibility under physiologically-relevant

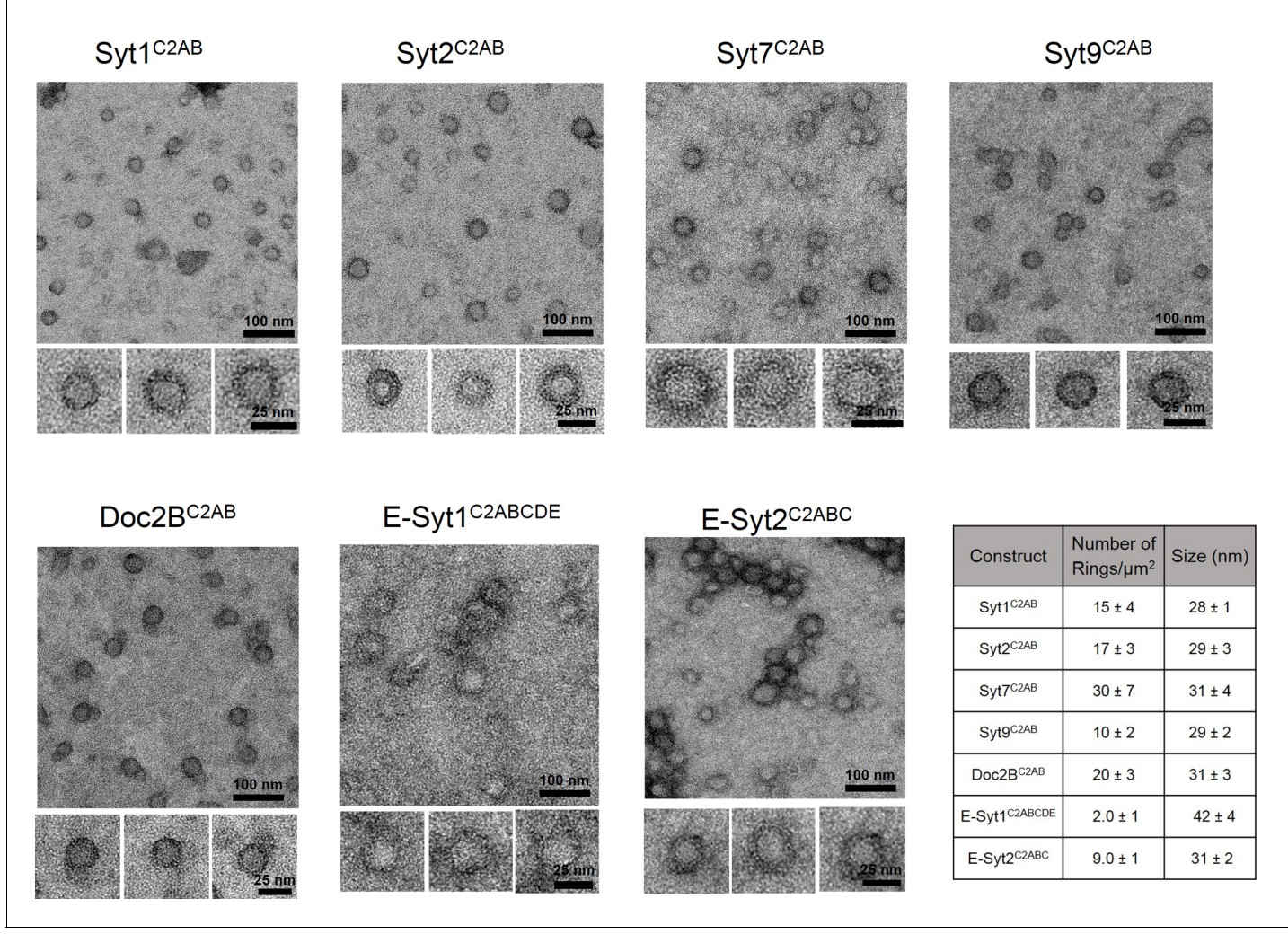

**Figure 1.** Ring-like oligomers are a common structural feature of C2 domain proteins. EM analysis showing the C2AB domains of neuronal isoforms of Syt, namely Syt1, Syt2, Syt7, and Syt9 form ring like oligomers on monolayers under $Ca^{2+}$-free conditions. Similar ring-like structures were observed for other related C2 domain proteins, like Doc2B and E-Syt 1 & 2. The number of ring-oligomers observed on the monolayers varied, but the dimensions of the rings were remarkably consistent (~30 nm). All EM analyses were carried out using 5 μM protein on monolayer containing 40% PS and buffer containing 15 mM KCl and 1 mM free $Mg^{2+}$. Representative micrographs and average values, along with standard error of the means (SEM) from a minimum of three independent trials are included.

The following figure supplements are available for figure 1:

**Figure supplement 1.** Ring assembly is not a conserved property of all C2 domains.

**Figure supplement 2.** Effect of $Ca^{2+}$ addition on pre-formed ring-like oligomers of Syt isoforms and other C2 domain proteins.

conditions. The ring oligomers assembled with the minimal C2AB domain of Syt1 were highly sensitive to the ionic strength of the buffer and the anionic lipid content on the monolayer. A minimum of 35% PS in the monolayer and buffers containing <50 mM KCl were required to obtain stable ring structures (*Wang et al., 2014*). We reasoned that the inclusion of conserved N-terminal juxtamembrane region (~60 residues) that connects the C2AB domains to the membrane anchor, might help stabilize the ring oligomers. The juxtamembrane linker domain has been shown to be vital for Syt1 role in activating synchronous release and in clamping the spontaneous release (*Caccin et al., 2015*; *Lee and Littleton, 2015*). It also has the ability to interact with the membrane and has been shown to self-oligomerize (*Fukuda et al., 2001*; *Lai et al., 2013*; *Lu et al., 2014*).

We purified the entire cytoplasmic domain of Syt1 (Syt1$^{CD}$, residues 83–421) using a stringent purification protocol (*Seven et al., 2013*; *Wang et al., 2014*) to remove all polyacidic contaminants, which could promote non-specific aggregation of the protein (*Seven et al., 2013*) and this is confirmed by a single peak in the size-exclusion chromatography (*Figure 2—figure supplement 1A*). As expected, lipid binding analysis showed that the juxtamembrane domain enhances and stabilizes the membrane interaction of Syt1 under physiologically-relevant experimental conditions (*Figure 2—figure supplement 1B*). To visualize the organization of the Syt1$^{CD}$ on lipid monolayers under Ca$^{2+}$-free conditions, we adapted the conditions used previously to obtain Syt1$^{C2AB}$ rings (*Wang et al., 2014*). Negative stain EM analysis showed that Syt1$^{CD}$ can form stable ring-like oligomers (*Figure 2A*) on monolayers under physiologically-relevant lipid (PC/PS at 3:1 molar ratio) and buffer (100 mM KCl, 1 mM free magnesium, Mg$^{2+}$) composition. The outer diameter of these Syt1$^{CD}$ rings ranged from 19–42 nm, with an average size of 30 ± 4.5 nm (*Figure 2B*), analogous to the Syt1$^{C2AB}$ rings (*Wang et al., 2014*). Based on the helical indexing of the Syt1$^{C2AB}$ tubes (*Wang et al., 2014*), we estimate that this corresponds to 12–25 copies of Syt1 molcule, with average ~17 copies of Syt1. The Syt1$^{CD}$ rings were robust as we did not observe many collapsed ring structures, like the 'clams' or 'volcanos', routinely seen with C2AB rings (*Wang et al., 2014*) and were stable under a wide-range of the ionic strengths and anionic lipid content (*Figure 2C*). Therefore, we used the Syt1$^{CD}$ to delineate the mechanistic details of the Syt1 ring oligomer assembly and its Ca$^{2+}$-sensitivity in a physiologically relevant environment.

## Syt1 C2B interaction with PIP2 is required for ring formation

The assembly of the Syt1$^{CD}$ ring oligomers strictly required the presence of anionic lipid (PS) in the monolayer (*Figure 2—figure supplement 2*) and the amount of the negative charge in the monolayer and the ionic strength of the buffer affected the number and integrity of the Syt1$^{CD}$ rings (*Figure 2C*). Therefore, to identify which parts of Syt1 are involved in positioning the Syt1 on the membrane to promote the ring assembly, we focused on the conserved polybasic regions of Syt1. Disrupting the polylysine motif on the C2A (K190A, K191A) or the arginine cluster on the C2B (R398A, R399A) did not affect the ring formation (*Figure 2D*), but mutations of key lysine residues (K326A, K327A) within the polybasic patch on the C2B drastically reduced (~90%) the number of the Syt1$^{CD}$ rings, even when 25% PS was included in the monolayer (*Figure 2D*). This suggests that the electrostatic interaction between the polylysine motif on C2B and the anionic lipids on the membrane surface is required for the ring formation.

Consequently, we tested the effect of PIP2 on the ring assembly as the polylysine motif on C2B has been shown to preferentially bind PIP2 with high affinity (*Bai et al., 2004*; *Parisotto et al., 2012*; *Park et al., 2012*; *Honigmann et al., 2013*; *Krishnakumar et al., 2013*; *Lai et al., 2015*). Syt1$^{CD}$ ring formation did not require PIP2, but inclusion of PIP2 in the lipid monolayer (25% PS, 3% PIP2, 72% PC) improved the number and the integrity of the Syt1$^{CD}$ rings (*Figure 3A and E*). However, PIP2 was essential to obtain stable Syt1$^{CD}$ ring oligomers when ATP at physiological concentrations (1 mM Mg-ATP) was included (*Figure 3B,C and E*). ATP is a critical co-factor, which modulates Syt1 function as it reverses the inactivating *cis-* interaction of Syt1 with its own membrane while preserving the functional *trans-* association to the plasma membrane (*Park et al., 2012*; *Vennekate et al., 2012*). This is because ATP effectively screens the interaction of Syt1 with weakly anionic PS, but not with the strong negative charges on the PIP2 head group found exclusively on the PM (*Park et al., 2012*, *2015*). Correspondingly, lipid binding assays showed that the ATP blocks the binding of Syt1$^{CD}$ to PS-containing vesicles, but not to PS/PIP2 membranes (*Figure 3—figure supplement 1*). Corroborating this, 6% PIP2 as the sole anionic lipid (6% PIP2, 94% PC) in the lipid monolayer was found to be sufficient to form ring oligomers, even in the presence of 1 mM ATP (*Figure 3D and E*). Taken together, our data shows that under physiological ionic conditions, the Ca$^{2+}$-independent interaction of the C2B domain with PIP2 on the PM, which has been implicated in the vesicle docking both in vitro and in vivo (*Wang et al., 2011*; *Parisotto et al., 2012*; *Park et al., 2012*; *Honigmann et al., 2013*; *Lai et al., 2015*), is key to assembling the Syt1 ring-like oligomers.

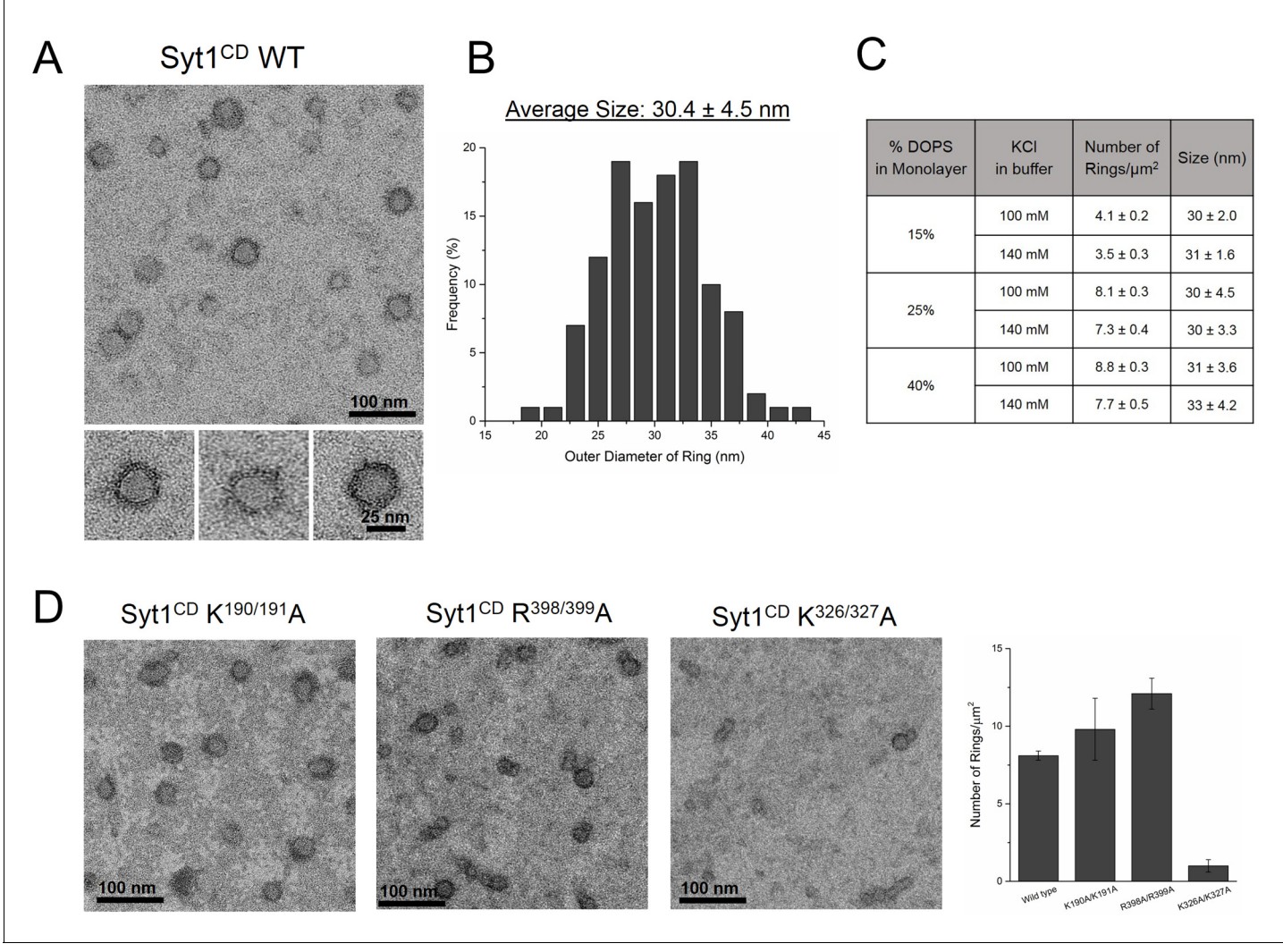

**Figure 2.** The entire cytoplasmic domain of Syt1 (Syt1$^{CD}$) forms ring-like oligomers under physiologically relevant conditions. (**A**) Negative stain EM analysis shows ring-like oligomers of Syt1$^{CD}$ on PC/PS (3:1 molar ratio) lipid monolayers in buffer containing 100 mM KCl and 1 mM MgCl$_2$. (**B**) The size distribution of the Syt1$^{CD}$ rings as measured from the outer diameter (n = ~400) under these experimental conditions using ImageJ software. (**C**) The Syt1$^{CD}$ ring-oligomers were observed under a wide-ranging conditions. Under all conditions tested, the dimension of these ring oligomers were very consistent (~30 nm), but the number of rings observed depended on amount of the anionic lipid in the monolayer and the salt (KCl) concentration of the buffer (**D**) EM analysis showing that the polylysine (K326/K327) motif of C2B domain is critical to the ring formation, but the other conserved polybasic regions of Syt1, namely K190/K191 on C2A and R398/R399 on C2B are not involved in ring formation. All EM analyses were carried out using 5 µM protein on monolayers containing 25% PS and in buffer containing 100 mM KCl and 1 mM free Mg$^{2+}$. Representative micrographs and averages/SEM from three independent trials are shown.

The following figure supplements are available for figure 2:

**Figure supplement 1.** Purification and Characterization of Syt1$^{CD}$.

**Figure supplement 2.** Presence of anionic lipid is required to assemble the Syt1$^{CD}$ ring oligomers.

## Ca$^{2+}$-triggered membrane insertion of Syt1 C2B disrupts the ring oligomers

Similar to Syt1$^{C2AB}$, Syt1$^{CD}$ rings were sensitive to Ca$^{2+}$ and brief treatment (~10 s) with Ca$^{2+}$ drastically disrupted the integrity of the preformed Syt1$^{CD}$ ring oligomers (**Figure 4A**). Calcium ions at concentrations in the range measured in intra-terminal region during synaptic transmission

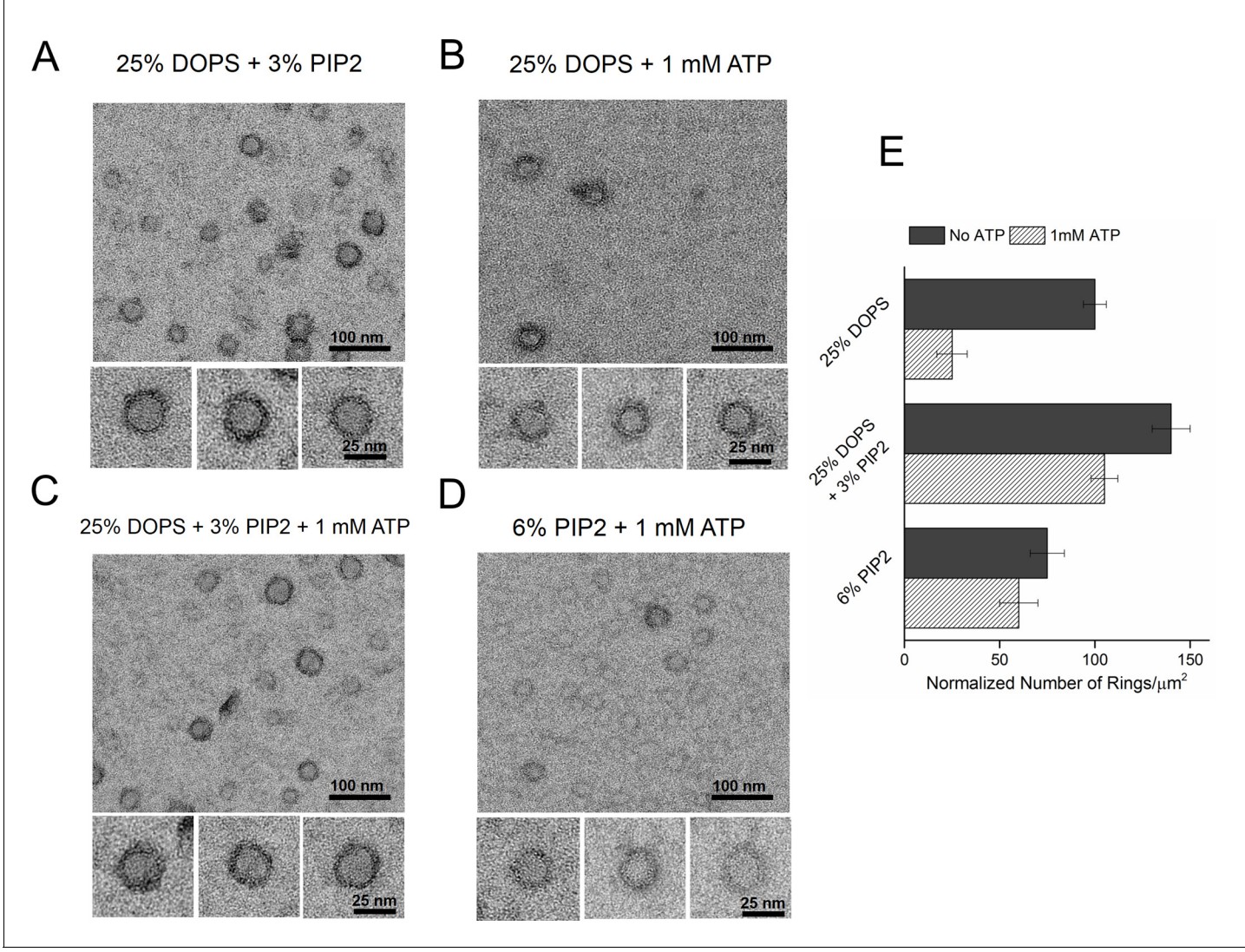

**Figure 3.** Syt1-PIP2 interaction is key to ring-formation under physiologically relevant conditions. (A) Inclusion of 3% PIP2 (in addition to 25% PS) in the monolayer stabilized the ring structures and increased the number of rings observed. (B, C) Addition of ATP drastically reduced the number of rings observed in monolayers containing 25% PS only, but not when supplemented with 3% PIP2. (D) PIP2 (6%) as the only anionic lipid on the bilayer was sufficient to assemble ring-like oligomers, even in the presence of 1 mM ATP. All EM analyses were carried out using 5 μM protein in buffer containing 100 mM KCl and 1 mM free Mg$^{2+}$. Representative micrographs and average values/SEM from a minimum of three independent trials are shown in (E). The rings observed under all conditions shown in (E) were similarly (~30 nm) sized

The following figure supplement is available for figure 3:

**Figure supplement 1.** Lipid binding analysis shows that ATP effectively screens the interaction of Syt1C$^D$ to PS-only membrane, but not membrane containing 3% PIP2.

(*Schneggenburger and Neher, 2000, 2005; Neher and Sakaba, 2008*) fragmented and disassembled the rings in a Ca$^{2+}$ concentration-dependent fashion (*Figure 4A*). PIP2 had little or no effect on the Ca$^{2+}$ sensitivity of the Syt1$^{CD}$ as we observed very similar reduction in Syt1$^{CD}$ rings with or without 3% PIP2 across all Ca$^{2+}$ concentration tested (*Figure 4—figure supplement 1*). To verify that the Ca$^{2+}$ sensitivity of the Syt1$^{CD}$ rings is indeed due to specific Ca$^{2+}$ binding to Syt1 and to map this sensitivity, we generated and tested Syt1$^{CD}$ mutants that disrupt Ca$^{2+}$ binding to the C2A and C2B domains respectively (*Shao et al., 1996*). As shown in *Figure 4B*, disrupting Ca$^{2+}$ binding to C2B (Syt1$^{CD}$ D309A, D363A, D365A; C2B$^{3A}$) rendered the ring oligomers insensitive to

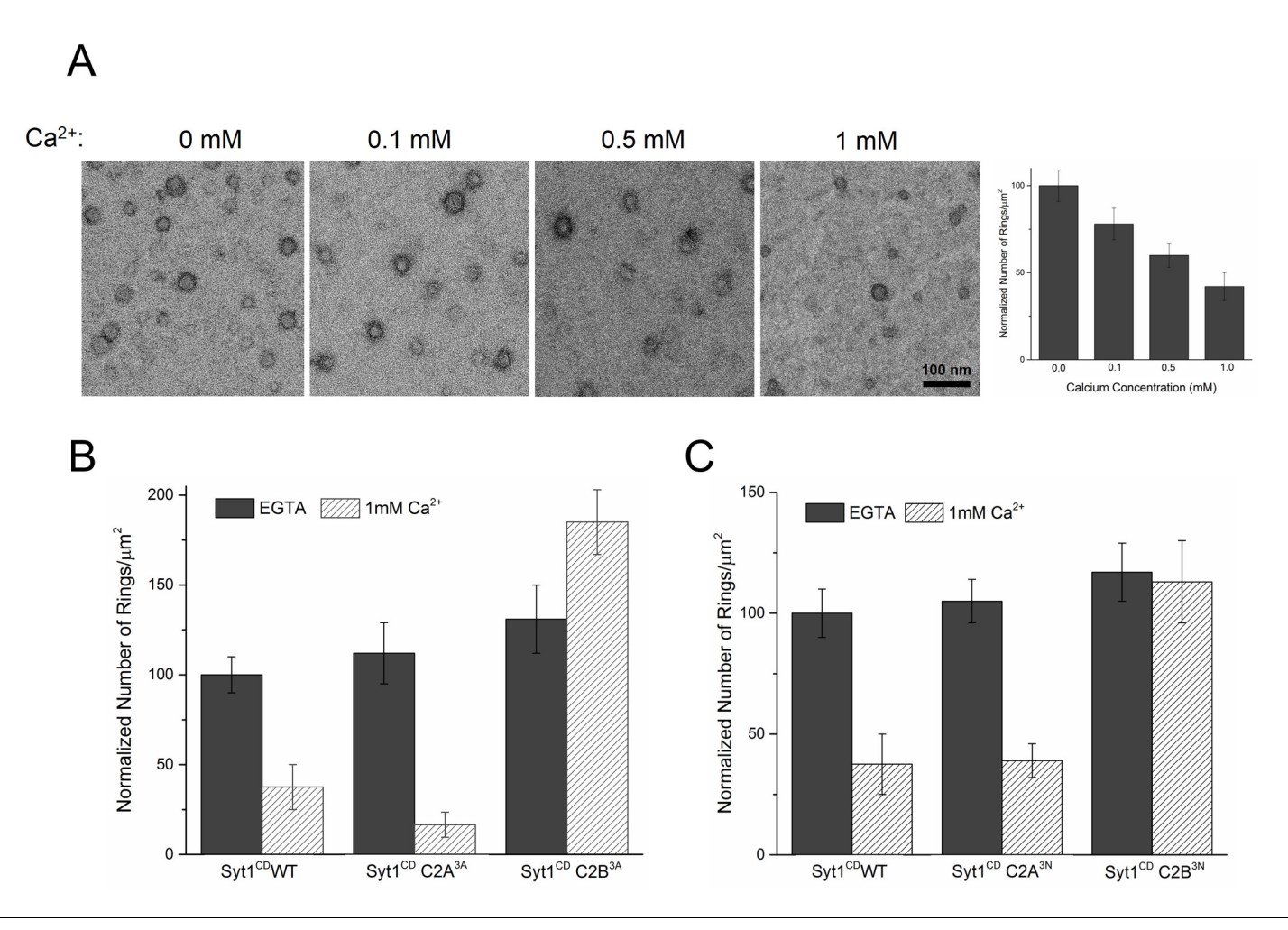

**Figure 4.** $Ca^{2+}$ binding and subsequent re-orientation of the C2B domain into the membrane are needed to disassemble the Syt1 ring oligomer. (**A**) $Syt1^{CD}$ ring oligomers were sensitive to $Ca^{2+}$ and brief treatment (10 s) of the pre-formed rings with physiological levels of $Ca^{2+}$ greatly reduced the number of rings observed. (**B**) $Ca^{2+}$ sensitivity of the $Syt1^{CD}$ rings maps to the C2B domain as disrupting $Ca^{2+}$ binding to C2B ($Syt1^{CD}$ D309A, D363A, D365A; $C2B^{3A}$) but not C2A ($Syt1^{CD}$ D178A, D230A, D232A; $C2A^{3A}$) rendered the rings $Ca^{2+}$ insensitive. (**C**) $Ca^{2+}$-induced insertion of just the C2B domain is necessary to disrupt the ring oligomers as hydrophilic mutation that blocks its insertion of the C2B loop ($Syt1^{CD}$ V304N, Y364N, I367N; $C2B^{3N}$) but not the C2A loop ($Syt1^{CD}$ F231N, F234N, S235N; $C2A^{3N}$) makes the rings insensitive to $Ca^{2+}$. All EM analyses were carried out using 5 µM protein on monolayers containing 25% PS and in buffer containing 100 mM KCl and 1 mM free $Mg^{2+}$. Effect of addition of 1mM $Ca^{2+}$ (final concentration) is shown in (B) & (C). Representative micrographs and average values and deviations (SEM) from 3–4 independent trials are shown.

The following figure supplements are available for figure 4:

**Figure supplement 1.** Inclusion of PIP2 does not change the $Ca^{2+}$ sensitivity of the $Syt1^{CD}$ ring oligomers.

**Figure supplement 2.** Disrupting the calcium binding to C2B ($Syt1^{CD}$ D309A, D363A, D365A; $C2B^{3A}$), but not C2A ($Syt1^{CD}$ D178A, D230A, D232A; $C2A^{3A}$) renders the $Syt1^{CD}$ rings insensitive to calcium.

**Figure supplement 3.** Disrupting the $Ca^{2+}$-induced membrane insertion of C2B loop ($Syt1^{CD}$ V304N, Y364N, I367N; $C2B^{3N}$) but not the C2A loop ($Syt1^{CD}$ F231N, F234N, S235N; $C2A^{3N}$) makes the rings insensitive to calcium.

calcium ions, while blocking $Ca^{2+}$ binding to the C2A domain ($Syt1^{CD}$ D178A, D230A, D232A; $C2A^{3A}$) did not alter the effect of $Ca^{2+}$ on the $Syt1^{CD}$ rings (**Figure 4—figure supplement 2**). Likewise, mutations of aliphatic loop residues in the C2B domain ($Syt1^{CD}$ V304N, Y364N, I367N; $C2B^{3N}$), which insert into the membrane following $Ca^{2+}$ binding, made the $Syt1^{CD}$ ring oligomers insensitive

to $Ca^{2+}$ wash, but corresponding mutations in the C2A calcium loops (Syt1[CD] F231N, F234N, S235N; C2A[3N]) had no effect (*Figure 4C*, *Figure 4—figure supplement 3*). The mutation analysis shows that the rapid disruption of the Syt1 rings requires $Ca^{2+}$ binding to the C2B and the subsequent reorientation of the C2B domain into the membrane. In other words, the dissociation of the Syt1 ring oligomers is coupled to the conformational changes in C2B domain, which is involved in $Ca^{2+}$ activation and is physiologically required for triggering synaptic transmission.

## Discussion

In support of a functional role for the Syt1 ring-oligomers, we find that the molecular basis of the Syt1 ring oligomer assembly and its reversal are coupled to well-established mechanisms of Syt1 action. The interaction of the conserved lysine residues in the polybasic region of the C2B domain with PIP2 on the inner leaflet of the pre-synaptic plasma membrane is a key determinant in both ring assembly and in synaptic vesicle docking *in vivo* (*Martin, 2012*; *Honigmann et al., 2013*), suggesting these processes are mechanistically linked. In addition, Syntaxin clusters PIP2 (by binding via its basic juxtamembrane region) and it has been suggested that it is these clusters that recruit the SVs (*Honigmann et al., 2013*). Given the high local concentration of both PIP2 (estimated to be up to ~80 mol% in such micro-domains [*Honigmann et al., 2013*]) and Syt1 (anchored in the synaptic vesicles), it is easy to imagine how the ring-like oligomers could form at the docking site in between the synaptic vesicle and the PM. There are ~16–22 copies of Syt1 on a synaptic vesicle (*Takamori et al., 2006*; *Wilhelm et al., 2014*), enough to form a ring oligomer of ~27–37 nm in diameter, assuming no contribution from the plasma membrane pool of Syt1. This is consistent with the Syt1 ring diameters observed on the lipid monolayers (*Figure 2B*). Several studies have shown that the Syt1-PIP2 docking interaction precedes the engagement of the v- with t-SNAREs (*van den Bogaart et al., 2011*; *Parisotto et al., 2012*). The prior formation of a Syt1 ring would thus position it to ideally prevent the complete zippering of the SNAREs, in addition to acting as a washer (or spacer) to separate the two membranes. The height of the ring, ~4 nm (*Wang et al., 2014*) would allow for the N-terminal domain of the SNARE complex to assemble, but such a gap would impede complete zippering. In effect, the Syt1 rings would block SNARE-mediated fusion and hold the SNARE in a pre-fusion half-zippered state (*Figure 5*). This is consistent with the earlier observation that docked vesicles appear to be 3–4 nm away from plasma membrane (*Fernandez-Busnadiego et al., 2011*).

Besides positioning the Syt1 to promote the ring assembly, the binding of the polybasic region to the PIP2 clusters on the PM would also hold back the $Ca^{2+}$ binding loops from the membrane (*Figure 5*). In fact, modeling of the C2AB domain onto the EM density map of the tubular structures of the Syt1[C2AB] suggests that the C2B calcium loops locates at the interface of the Syt1 oligomer (*Figure 5—figure supplement 1*). Such an arrangement would explain how the Syt1 ring could synchronize SV fusion to $Ca^{2+}$ influx. $Ca^{2+}$ binding to the C2B domain and subsequent conformational change, which incidentally is required to trigger neurotransmitter release (*Fernández-Chacón et al., 2001*; *Rhee et al., 2005*; *Paddock et al., 2011*), would induce reorientation of the C2B domain from the ring geometry and thus, break the ring oligomers. As such, this would remove the steric barrier and permit the stalled SNAREpins to complete zippering and trigger SV fusion to release neurotransmitters (*Figure 5*). This is congruent with the recent report (*Bai et al., 2016*), showing that the switch between the functional states (clamped vs. activated) of Syt1 involves large conformational change in the C2 domains.

Besides membranes, Syt1 also binds to t-SNAREs and this interaction is functionally relevant for fast neurotransmitter release (*de Wit et al., 2009*; *Mohrmann et al., 2013*; *Zhou et al., 2015*; *Wang et al., 2016*). Recent reports have mapped the key t-SNARE binding interface to the C2B domain (*Zhou et al., 2015*), which is believed to form before the influx of $Ca^{2+}$ and is maintained during $Ca^{2+}$ activation process (*Krishnakumar et al., 2013*; *Zhou et al., 2015*; *Wang et al., 2016*). We note that in our Syt1 ring oligomer model, this binding interface on the C2B (*Figure 5—figure supplement 1*) is accessible and free to interact with the SNAREs. However, the occupancy and positioning of the SNARE complexes on the Syt1 ring oligomer is not known and as such, is the focus of our ongoing research. Nevertheless, it is easy to imagine that such an interaction would allow the Syt1 ring to act as a primer to organize the core components of the fusion machinery to allow for a rapid and synchronous neurotransmitter release. Further, the oligomeric structure could provide a

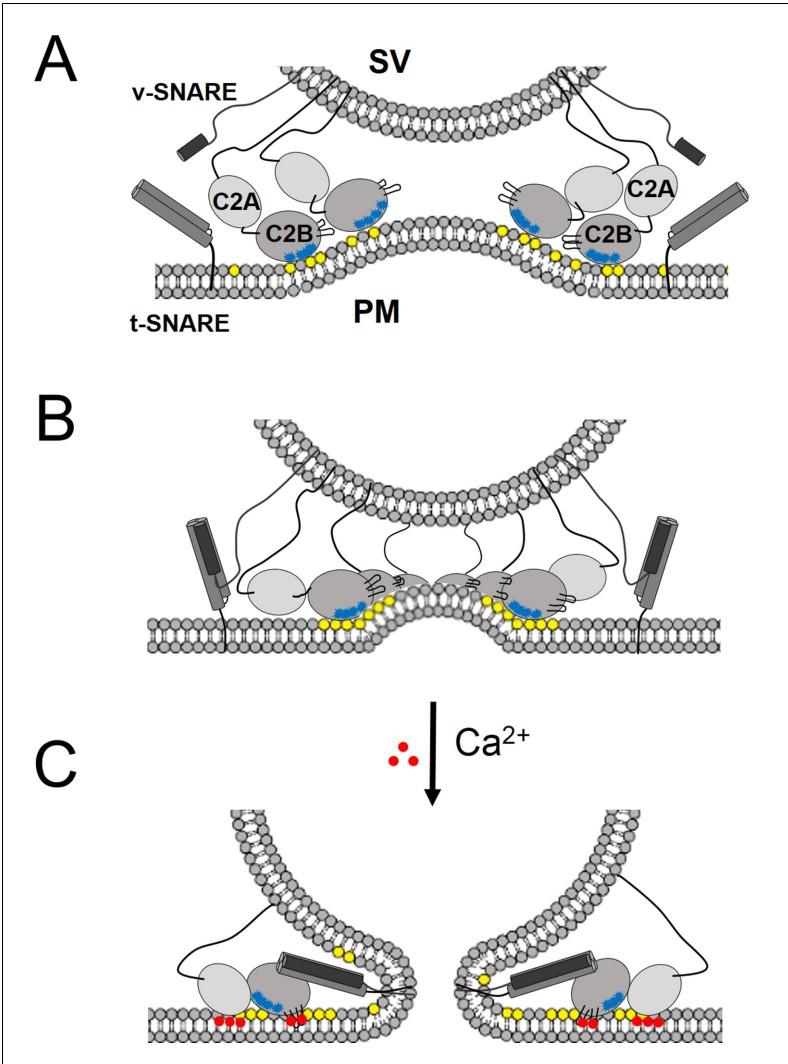

**Figure 5.** 'Washer' model for the regulation of neurotransmitter release by Syt1. (**A**) The SV docking interaction of the Syt1 polylysine motif (blue dots) with the PIP2 (yellow dots) on the plasma membrane positions the Syt1 on the membrane to promote the ring-oligomer formation. The ring assembly might precede the engagement of the SNARE proteins. (**B**) Syt1 ring-oligomers assembled at the SV-PM interface act as a spacer or 'washer' to separate the two membranes. The height of the ring (~4 nm) would allow the partial assembly of the SNARE complex, but prevent complete zippering and thus, block fusion. NOTE: The positioning and occupancy of SNAREs on Syt1 ring is not known and are shown for illustrative purposes only. (**C**) Upon binding calcium ions (red dots), the $Ca^{2+}$ loops that locates to the oligomeric interface, re-orients and inserts into the membrane, thus disrupting the ring oligomer to trigger fusion and release neurotransmitters. Thereby, the Syt1 ring oligomers will synchronize the release neurotransmitters to the influx of calcium ions.

The following figure supplement is available for figure 5:

**Figure supplement 1.** Organization of Syt1 C2 domains in the ring oligomers.

mechanistic basis for the observed $Ca^{2+}$-cooperativity in triggering SV fusion. Obviously, the 'washer' model is speculative and functional and physiological studies are required to ascertain its relevance.

Based on our data, the key principles of the ring oligomer assembly and its $Ca^{2+}$ sensitivity can be summarized as follows: The ring-oligomer formation is mediated by a single C2 domain (within multi-C2 domain protein), which binds the anionic lipids on membrane surface via the polybasic

motif (*Figures 1* and *2*) and $Ca^{2+}$ induced re-orientation of the same C2 domain away from the ring geometry disrupts the ring oligomers (*Figure 4*). In other words, the $Ca^{2+}$ sensitivity of the ring oligomers requires the same C2 domain to have the capacity to bind both anionic lipids and $Ca^{2+}$. This is true for the C2AB domains of the Syt isoforms and Doc2B and hence, these ring oligomers are $Ca^{2+}$ sensitive (*Figure 1—figure supplement 2*). However, in the case of the E-Syts, the C-terminal C2 domains (C2E for E-Syt1 and C2C in E-Syt1) that are involved in anionic lipid dependent membrane tethering (thereby the ring formation) lack the putative $Ca^{2+}$ binding loops, with the N-terminal C2 domains mediating the $Ca^{2+}$-dependent membrane interaction (*Giordano et al., 2013*; *Reinisch and De Camilli, 2016*). Hence, the E-Syt rings are insensitive to $Ca^{2+}$ (*Figure 1—figure supplement 2*). Further, E-Syt1 exhibits very weak membrane binding under $Ca^{2+}$ free conditions, which is enhanced upon $Ca^{2+}$ addition (*Idevall-Hagren et al., 2015*). The increased surface concentration of the E-Syt1 in the presence of $Ca^{2+}$ could explain the improvement in the number of E-Syt1 rings observed under these conditions (*Figure 1—figure supplement 2*).

In summary, we find that ring-like oligomers are a common structural feature of C2 domain containing proteins, not all of which are regulators of exocytosis. Particularly interesting are the E-Syts, which function to enable the ER and plasma membrane to come into intimate contact – close enough for lipids to be transferred. Our results suggest this might be achieved by bridging two membranes with an intervening structure, most probably based on ring oligomers. Such an organization could stabilize the contact sites and also enhance the lipid transfer function of E-Syts. However, more research is required to understand this better. Interestingly, yeast cells have both E-Syts (for membrane adhesion) and SNAREs (for membrane fusion) but do not contain vesicle-associated Syt protein and do not carry out calcium-regulated exocytosis. Perhaps this set the stage for exocytosis to evolve when the C2 domains combined with a vesicle-associated protein to form ring-like oligomer i.e. washers that reversibly impeded SNAREpins.

## Materials and methods

The DNA constructs used in this study are the rat synaptotagmin-1 C2A and C2B domain (Syt1$^{C2AB}$, residues 143–421); entire cytoplasmic domain (Syt1$^{CD}$, residues 83–421); human synaptotagmin-2 C2A and C2B domain (Syt2$^{C2AB}$, residues 141–419); human synaptotagmin-7 C2A and C2B domain (Syt7$^{C2AB}$, residues 130–404); human synaptotagmin-9 C2A and C2B domain (Syt9$^{C2AB}$ residues 222–491), mouse Doc2B C2A and C2B domain (Doc2B$^{C2AB}$ residues 128–412); human extended syanptotagmin-1 C2A, C2B, C2C, C2D and C2E domain (E-Syt1$^{ABCDE}$, residues 315–1104), human extended syanptotagmin-2 C2A, C2B and C2C domain (E-Syt2$^{ABC}$, residues 351–893). The following mutants in Syt1$^{CD}$ background was created using QuikChange mutagenesis kit (Agilent Technologies, Santa Clara, CA): $Ca^{2+}$-binding mutant in C2A (C2A$^{3A}$, SYT1 residues 83–421 with D$^{178}$A, D$^{230}$A, D$^{232}$A) and in C2B (C2B$^{3A}$, D$^{309}$A, D$^{363}$A, D$^{365}$A), the calcium loop insertion mutant on C2A (C2A$^{3N}$, F$^{231}$N, F$^{234}$N, S$^{235}$N) and C2B (C2B$^{3N}$, V$^{304}$N, Y$^{364}$N, I$^{367}$N), the lysine patch mutation in C2A (K$^{190}$A,K$^{191}$A), in C2B (K$^{326}$A, K$^{327}$A) and arginine patch mutation in C2B (R$^{398}$A, R$^{399}$A). Lipids, 1,2-dioleoyl-*sn*-glycero-3-phosphocholine (DOPC), and 1,2-dioleoyl-sn-glycero-3-phospho-L-serine (DOPS), phosphatidylinositol 4, 5-bisphosphate (PIP2), were purchased from Avanti Polar Lipids (Alabaster, AL).

### Protein expression and purification

The Syt1$^{CD}$ wild-type and mutant proteins were expressed and purified as a His$^6$-tagged protein using a pET28 vector, while Syt$^{C2AB}$ isoforms and Doc2B were expressed and purified as a GST-construct. The proteins were purified as described previously (*Seven et al., 2013*; *Wang et al., 2014*), with few modifications. Briefly, Escherichia coli BL21 (DE3) expressing Sytconstructs were grown to an $OD_{600}$ ~0.7–0.8, induced with 0.5 mM isopropyl β-D-1-thiogalactopyranoside (IPTG). The cells were harvested after 3 hr at 37°C and suspended in lysis buffer (25 mM HEPES, pH 7.4, 400 mM KCl, 1 mM $MgCl_2$, 0.5 mM TCEP, 4% Triton X-100, protease inhibitors). The samples were lysed using cell disrupter, and the lysate was supplemented with 0.1% polyethylimine before being clarified by centrifugation (100,000 ×g for 30 min). The supernatant was loaded onto Ni-NTA (Qiagen, Valencia, CA), or Glutathione-Sepharose (Thermo Fisher Scientific , Grant Island, NY) beads (3 hr or overnight at 4°C) and the beads was washed with 20 ml of lysis buffer, followed by 20 ml of 25 mM HEPES, 400 mM KCl buffer containing with 2 mM ATP, 10 mM $MgSO_4$, 0.5 mM TCEP. Subsequently, the beads were resuspended in 5 ml of lysis buffer supplemented with 10 μg/mL DNaseI,

10 µg/mL RNaseA, and 10 µl of benzonase (2000 units) and incubated at room temperature for 1 hr, followed by quick rinse with 10 ml of high salt buffer (25 mM HEPES, 1.1 M KCl, 0.5 mM TCEP) to remove the nucleotide contamination. The beads were then washed with 20 ml of HEPES, 400 mM KCl buffer containing 0.5 mM EGTA to remove any trace calcium ions. The proteins were eluted off the affinity beads in 25 mM HEPES, 100 mM KCl, 0.5 mM TCEP buffer, either with 250 mM Imidazole (His-tag proteins) or using Precission protease for GST-tagged constructs and further purified by anionic exchange (Mono-S) chromatography. Size-exclusion chromatography (Superdex75 10/300 GL) showed a single elution peak (~12 mL) consistent with a pure protein, devoid of any contaminants.

Coding sequences of C2A-E domains from human E-Syt1 was cloned into pCMV6-AN-His vector (OriGene). The plasmid was transfected into Expi293 cells (Thermo Fisher Scientific, Grant Island, NY) for protein expression. After three days of transfection, cells were collected and lysed by three cycles of freeze and thaw (liquid $N_2$ and 37°C water bath). His-tagged E-ESyt1$^{C2ABCDE}$ was then purified by His$^{60}$ Nickel Resin (Clontech, Mountain View, CA), with Imidazole elution. For E-Syt2$^{ABC}$ production, the coding sequence was cloned into a modified pCDFDuet-1 vector (Novagen, Danvers, MA), which has an N-terminal GST tag and a Prescission protease cleavage site and transformed into BL21(DE3). The cells were grown at 37°C to an $OD_{600}$ of ~0.6–0.8, then were shifted to 22°C before induction with 0.5 mM IPTG. Cells were harvested 18 hr after induction. The proteins were purified by Glutathione Sepharose 4B chromatography . GST tags were removed by treatment with Prescission protease. Both E-Syt proteins were further purified by gel filtration on a Superdex200 column . The gel filtration buffer contained 20 mM HEPES at pH 8.0, 150 mM NaCl, and 0.5 mM TCEP. All chromatrography was carried out using AKTA system (GE Healthcare, Marlborough, MA)

In all cases, the protein concentration was estimated using Bradford assay with BSA as standard and the nucleotide contamination was tracked using the 260 nm/280 nm ratios. The protein was flash frozen and stored at −80°C with 10% glycerol (20% glycerol for Syt1$^{CD}$) without significant loss of ring-forming activity.

## Lipid monolayer assay

To form the lipid monolayer, degased ultrapure $H_2O$ was injected through a side port to fill up wells (4 mm diameter, 0.3 mm depth) in a Teflon block. The surface of the droplet was coated with 0.5 µl of phospholipid mixture (0.5 mM total lipids). The lipid mixtures, DOPC/DOPS & DOPPC/DOPS/PIP2 were pre-mixed as required, dried under $N_2$ gas and then re-suspended in chloroform to the requisite concentration before adding to the water droplet. The Teflon block then was sealed in a humidity chamber for 1 hr at room temperature to allow the chloroform to evaporate. Continuous carbon-coated EM grids (400 mesh; Ted Pella Inc., Redding, CA ) were baked at 70°C for 1 hr and washed with hexane to improve hydrophobicity. Lipid monolayers formed at the air/water interface were then recovered by placing the pre-treated EM grid carbon side down on top of each water droplet for 1 min. The grid was raised above the surface of the Teflon block by injecting ultrapure $H_2O$ into the side port and then was lifted off the droplet immediately.

Proteins were rapidly diluted to 5 µM in 20 mM MOPS, pH 7.5, 5 mM KCl, 1 mM EDTA, 2 mM MgAC$_2$, 1 mM DTT, 5% (wt/vol) trehalose buffer and then added to the lipid monolayer on the grid and incubated in a 37°C humidity chamber for 1 min. The final KCl concentration in the buffer were adjusted to 100 mM or 140 mM as required. To facilitate structural analysis of the rings, we further optimized the incubation conditions by using an annealing procedure: Rings were nucleated at 37°C for 1 min followed by a 30-min annealing step at 4°C. The grids were rinsed briefly (~10 s) with incubation buffer alone or with buffer supplemented with CaCl$_2$ (0.1, 0.5 and 1 mM free) for Ca$^{2+}$ treatment studies. The free [Ca$^{2+}$] was calculated by Maxchelator (maxchelator.stanford. edu). Subsequently, the grids were blotted with Whatman#1 filter paper (Sigma-Aldrich, St. Louis, MO), negatively stained with uranyl acetate solution (1% wt/vol), and air dried. The negatively stained specimens were examined on a FEI Tecani T12 operated at 120 kV. The defocus range used for our data was 0.6–2.0 µm. Images were recorded under low-dose conditions (~20 e−/Å2) on a 4K × 4K CCD camera (UltraScan 4000; Gatan, Inc., Pleasanton, CA), at a nominal magnification of 42,000×. Micrographs were binned by a factor of 2 at a final sampling of 5.6 Å per pixel on the object scale. The image analysis, including size distribution measurements was carried out using ImageJ software.

## Acknowledgements

We thank Dr. Pietro De Camilli for helpful discussions and critical reading of the manuscript. We also wish to thank Dr. Kirill Volynski and Dr. Sarah Auclair for critical reading of the manuscript. This work was supported by National Institute of Health grant GM071458 to JER.

## Additional information

### Funding

| Funder | Grant reference number | Author |
|---|---|---|
| National Institute of General Medical Sciences | GM071458 | James E Rothman |

The funders had no role in study design, data collection and interpretation, or the decision to submit the work for publication.

### Author contributions

MNZ, ODB, Conception and design, Acquisition of data, Analysis and interpretation of data, Drafting or revising the article; JW, Conception and design, Acquisition of data, Analysis and interpretation of data; JC, YC, Conception and design, Drafting or revising the article, Contributed unpublished essential data or reagents; CVS, JER, SSK, Conception and design, Analysis and interpretation of data, Drafting or revising the article

### Author ORCIDs

James E Rothman, ⃝iD http://orcid.org/0000-0001-8653-8650
Shyam S Krishnakumar, ⃝iD http://orcid.org/0000-0001-6148-3251

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
