## [Decision Letter]

Thank you for submitting your article "Ring-like Oligomers of Synaptotagmins and Related Proteins" for consideration by *eLife*. Your article has been reviewed by three peer reviewers, and the evaluation has been overseen by a Reviewing Editor and Randy Schekman as the Senior Editor. One of the two reviewers has agreed to reveal his identity: Edwin R Chapman (Reviewer #2).

The reviewers have discussed the reviews with one another and the Reviewing Editor has drafted this decision to help you prepare a revised submission.

Summary:

This work extends a previous study (Wang et al. 2014, PNAS) on ring-like structures of synaptotagmin 1 on a monolayer. The most interesting point throughout the work is that the ring-formation ability seems to be shared for some, but not all, C2 domain-bearing proteins involved in membrane traffic, including certain E-Syts. However, there are several concerns that need to be addressed before a final decision can be made.

Essential revisions:

1) It is interesting that Syt1 C2AB/C2B could form ring-like oligomers in the presence of acidic lipids-bearing monolayer. However, it is unclear whether the C2 domain could self-assemble to oligomers in solution (i.e. in the absence of membrane), especially for Syt1^CD^ (residues 82-421). As the authors mentioned, the Syt1^CD^ is prone to aggregate (Fukuda et al. 2001, Lai et al. 2013, Lu et al. 2014) into oligomers. Note that Seven et al. 2015, PNAS suggest that the aggregation might due to impurity of the protein. Although this work used a developed method to purify the Syt1^CD^, the elution profile of Syt1^CD^ (with or without Mg^2+^ ions and calcium, also the other C2 domain proteins) on SEC should be provided.

2) Stable Syt1^CD^ ring oligomers were observed on membranes (with PIP2) in the presence of Mg^2+^, ATP and 100 mM KCl (Figure 3). Do Syt1 C2AB and/or C2B behave similarly in this condition? In the proposed model (Figure 5—figure supplement 1), Syt1 associates with each other via C2B-C2B binding interface but not via the juxtamembrane region that connects C2A and the transmembrane domain.

3) In Figure 1, the authors presented that a wide range of C2 domain-bearing proteins are able to form ring structures. However, in physiological conditions, some of these proteins may coexist in the same compartment (e.g. Syt1, Syt7 and Doc2). As the authors mentioned, these three proteins mediate different modes of neurotransmitter release separately and all can form similar ring structures. Can different C2 domains form hetero-oligomeric rings?

4) As shown in Figure 1 and Figure 1—figure supplement 2, E-Syt1 displayed very few and unstable ring structures in the absence of calcium, and surprisingly, such structures can be stabilized in the presence of calcium. This result is conflict with those observed for Syt1, 2, 7, 9 and Doc2B, suggesting E-Syt1 or 2 uses a different mechanism to form ring-like oligomers. Instead, the authors attributed these differences to 1): "The lack of ring-like oligomers for E-Syt1 might be due to the insufficient concentration of this protein on the membrane surface as E-Syt1 has very weak affinity to the membrane under calcium-free conditions (Idevall-Hagren et al. 2015)"; 2) the sensitivity to calcium is divergent between Syts/Doc2B and E-Syts; and "in fact, calcium improved the number of rings observed with E-Syt1 consistent with the fact effective membrane interaction of E-Syt1 requires calcium” (Figure 1—figure supplement 2 legend). However, how can one explain that calcium disrupted the number of rings observed with Syts and Doc2B considering that their binding to membrane is effective in the presence of calcium? The authors should check whether E-Syt1 bind to PS/PIP2 via the conserved polybasic region, and whether the binding of E-Syt1 to membranes is sensitivity to calcium.

5) In Figure 4 and Figure 4—figure supplement 1, the ring structures were disrupted after calcium addition. However, previous works show that there is a synergy between Syt1 (K326, K327)- PIP2 interaction and Syt1 (calcium-binding loops)-PS interaction (van den Bogaart et al. 2012, JBC). Moreover, the interaction between polybasic stretch (K326, K327) of Syt1 C2B and PIP2 has been reported not to be disrupted during calcium addition in the presence of t-SNARE/cis-SNARE complex (Wang et al. 2016, *eLife*). It is sometimes possible that the negative stain process affects the sample. Therefore, it is essential to reproduce the results with cryo-EM for samples treated by calcium in a short time before vitrification, to confirm whether Syt1 proteins stand around the disappearing ring structures similar to that observed with Syt2 in negative stain (Figure 1—figure supplement 2).

6) In the Discussion, the authors mentioned that "However, the molecular details of this interaction remains unclear, specifically due to the promiscuity and the low affinity of the Syt1/t-SNARE interaction". In recent work (Wang et al. 2016, *eLife*), it has been shown that the disassociation constant (Kd) between Syt1 and SNARE complex is about 1 μM in the presence of acidic membrane, compared to a large value obtained in the absence of membrane. The authors should cite this when discuss the functional relevant issues between Syt1 and t-SNARE.

7) Zhou et al. (Nature, 2015), showed that one of the interfaces between the synaptogtamin C2B domain and the SNARE complex is essential for fast neurotransmitter release and forms prior to calcium injection. Are the proposed models of Syt1 oligomers compatible with this finding? The authors are kindly encouraged to provide a balanced discussion of the recent findings by Wang et al., 2016 and Zhou et al. 2015 and discuss their models in the context of these recent findings.

8) The ring-oligomers formed by the E-Syts were not disrupted by calcium what are the implications?

9) In the legend for Figure 1—figure supplement 2, the authors mention that: "In fact, calcium improved the number of rings observed with E-Syt1 consistent with the fact effective membrane interaction of E-Syt1 requires calcium (Ref)." It is unclear as to what is meant here, since calcium stimulates the binding of Syts and Doc2 to membranes, as well.

10) It is unclear how many syt molecules are required to form the ring-like oligomers: 15? More than 15? What is the variation? What is the relevance to the copy number of syt1 per synaptic vesicle? The number of syt1 molecules per vesicle should be discussed, and (Takamori et al. 2006) cited. If it the required number is more than 15, it is highly unlikely that such rings form in vivo considering the protein number in synaptic vesicles. Moreover, given the physical constraints in the system (length of the linker that connects the C2-domains to the SV membrane etc.), more through and balanced discussion over whether syt1 can physically form these structures is required.

11) Given the physical constraints in vivo (e.g. membrane tethers, the diameter of synaptic vesicles etc.), can these rings form in vivo? And it might be wise to reexamine how the SNARE fit in, as they would seem to fit INSIDE these very large rings?

12) A large conformational switch in syt1 was recently reported (e.g. Bai et al., 2016), and this report seems congruent with the model put forward in the current study, so this paper should be discussed and cited.

13) Complexin as a fusion clamp – the new review from Trimbuch & Rosenmund (2016) should be cited, as there is strong evidence complexin is not a clamp in murine neurons.

14) The observation of E-Syt rings is very interesting. The lipid transfer activities of E-Syts remain intact when their C2 domains are replaced by those of Syt-1, consistent with the authors' model. However, the lipid-harboring synaptotagmin-like-mitochondrial-lipid binding protein (SMP) domains were not included in this study. Are these SMP domains compatible with the configuration of the ring if they all face inside?

15) Does PIP2 influence the ring-forming properties of other C2 domain molecules such as Doc2b and E-Syts?

16) The authors propose that the E-Syt rings prevent membrane fusion at ER-PM contact sites. This seems unlikely because membrane fusion is not a spontaneous process that needs to be "prevented". Instead, the ER-PM contact sites may simply lack the membrane fusion machinery. In support of this notion, lipid transfer also occurs at other membrane contact sites where no C2 domain molecules are found. These concepts need to be addressed by the authors.

17) Generally, the results with E-Syt are among the most interesting new results in this paper, but yet the discussion almost exclusively focuses on Syt1 (which is more controversial in view of the copy number in synaptic vesicles and other concerns raised in the above points). Thus, the authors are kindly asked to expand the discussion of the E-Syt results and their biological implications.

Optional revisions:

18) As Honigmann et al. 2013, NSMB and van den Bogaart et al. 2011, Nature reported, PIP2 displays a roughly 73 nm-sized microdomains in company with syntaxin-1a. In this work, however, there is no syntaxin reconstituted on monolayer. It is conceivable that Syt1 binds preferentially to PIP2 rather than other acidic lipids (e.g. PS) in the absence of Ca^2+^. How can the ~30 nm-sized ring structure be stabilized? What is the morphology of the ring structure in the presence of syntaxin and PIP2 without calcium?

19) Are the ring-like structures of Syt-1 affected by complexin and/or t-SNAREs (e.g., anchored to monolayers through Nickel-His6 interactions)?

---

## [Author Response]

*This work extends a previous study (Wang et al. 2014, PNAS) on ring-like structures of synaptotagmin 1 on a monolayer. The most interesting point throughout the work is that the ring-formation ability seems to be shared for some, but not all, C2 domain-bearing proteins involved in membrane traffic, including certain E-Syts. However, there are several concerns that need to be addressed before a final decision can be made.*

Essential revisions:

1) It is interesting that Syt1 C2AB/C2B could form ring-like oligomers in the presence of acidic lipids-bearing monolayer. However, it is unclear whether the C2 domain could self-assemble to oligomers in solution (i.e. in the absence of membrane), especially for Syt1^CD^ (residues 82-421). As the authors mentioned, the Syt1^CD^ is prone to aggregate (Fukuda et al. 2001, Lai et al. 2013, Lu et al. 2014) into oligomers. Note that Seven et al. 2015, PNAS suggest that the aggregation might due to impurity of the protein. Although this work used a developed method to purify the Syt1^CD^, the elution profile of Syt1^CD^ (with or without Mg^2+^ ions and calcium, also the other C2 domain proteins) on SEC should be provided.

As highlighted by the reviewer, we have used a stringent purification protocol to remove all polyacidic (nucleotide) contaminants, which could promote protein aggregation. This involved benzonase/ DNAse/ RNAse treatment, high salt (1.1 M KCl) wash and final purification on an ion exchange (Mono-S) column. The 260/280 ratio for all Syt1^CD^ samples after Mono-S purification were <0.55, confirming the purity of samples (NOTE: 260/280 ratio of 0.57 or less indicates no nucleotide contamination). Few selected samples, we further tested by size-exclusion chromatography (Superdex-75 column), which showed a single elution peak (~12 mL) consistent with a pure protein, devoid of any contaminants. As suggested by the reviewer, we now cite Seven et al.(Seven et al. 2013) and discuss the contaminant issue in detail in the Results (subsection “Complete Cytoplasmic Domain of Syt1 forms Rings under Physiologically Relevant Conditions”, last paragraph) and Methods (subsection “Protein Expression and Purification”, first paragraph) and also included the SEC profile (Figure 2—figure supplement 1). We tested the C2AB domain of the other Syt isoforms and Doc2B (i.e. without the juxtamembrane linker domain) and the aggregation is not a problem typically associated with the C2AB domain alone. Nevertheless, we employed the stringent method to purify the proteins. Thus, we believe that the oligomerization we have described on the lipid monolayers is an intrinsic and conserved feature of the C2 domain and not an aggregation triggered by impurities in the sample.

2) Stable Syt1^CD^ ring oligomers were observed on membranes (with PIP2) in the presence of Mg^2+^, ATP and 100 mM KCl (Figure 3). Do Syt1 C2AB and/or C2B behave similarly in this condition? In the proposed model (Figure 5—figure supplement 1), Syt1 associates with each other via C2B-C2B binding interface but not via the juxtamembrane region that connects C2A and the transmembrane domain.

Syt1^C2AB^ does form ring-like structures under more physiological conditions, but the prevalence and stability of the ring structures are drastically reduced (Wang et al. 2014. Figure S2). This, we beleive, is mainly due to the low concentration of the protein on the lipid monolayer surface under these conditions as we can increase the number of rings by increasing the negative lipid content on the monolayer and/or by lowering the salt concentration in the buffer to minimize the electrostatic screening effect. Lipid binding analysis (Figure 2—figure supplement 1) shows that the juxtamembrane linker domain indeed enhances and stabilizes the Syt1 membrane interaction under physiological conditions. Thus, inclusion of the juxtamembrane linker domain increases the surface concentration of Syt1 under these experimental conditions and thereby the prevalence of the ring oligomers. This data is now included in the revised manuscript (Figure 2—figure supplement 1) and referred in the Results section (subsection “Complete Cytoplasmic Domain of Syt1 forms Rings under Physiologically Relevant Conditions”, last paragraph).

3) In Figure 1, the authors presented that a wide range of C2 domain-bearing proteins are able to form ring structures. However, in physiological conditions, some of these proteins may coexist in the same compartment (e.g. Syt1, Syt7 and Doc2). As the authors mentioned, these three proteins mediate different modes of neurotransmitter release separately and all can form similar ring structures. Can different C2 domains form hetero-oligomeric rings?

Our data suggests that the circular oligomerization is an intrinsic property of the certain C2 domains, including many Syt isoforms and Doc2B. However, we have not explicitly tested the ability of the different isoform to form hetero-oligomeric ring structures. This is a very interesting possibility, with wide-ranging implication and will be focus of our future work and as such, is beyond the scope of the current work.

4) As shown in Figure 1 and Figure 1—figure supplement 2, E-Syt1 displayed very few and unstable ring structures in the absence of calcium, and surprisingly, such structures can be stabilized in the presence of calcium. This result is conflict with those observed for Syt1, 2, 7, 9 and Doc2B, suggesting E-Syt1 or 2 uses a different mechanism to form ring-like oligomers. Instead, the authors attributed these differences to 1): "The lack of ring-like oligomers for E-Syt1 might be due to the insufficient concentration of this protein on the membrane surface as E-Syt1 has very weak affinity to the membrane under calcium-free conditions (Idevall-Hagren et al. 2015)"; 2) the sensitivity to calcium is divergent between Syts/Doc2B and E-Syts; and "in fact, calcium improved the number of rings observed with E-Syt1 consistent with the fact effective membrane interaction of E-Syt1 requires calcium” (Figure 1—figure supplement 2 legend). However, how can one explain that calcium disrupted the number of rings observed with Syts and Doc2B considering that their binding to membrane is effective in the presence of calcium? The authors should check whether E-Syt1 bind to PS/PIP2 via the conserved polybasic region, and whether the binding of E-Syt1 to membranes is sensitivity to calcium.

We agree that the divergence in the Ca^2+^ sensitivity of the ring oligomers was not adequately addressed, so we have now included the following paragraph in the Discussion section to address this issue and also refer to in the Results section (subsection “Circular Oligomeric Assembly is a Common Feature of C2 Domain Proteins”): “Based on our data, the key principles of the ring oligomer assembly and its Ca^2+^ sensitivity can be summarized as follows: The ring-oligomer formation is mediated by a single C2 domain (within multi-C2 domain protein), which binds the anionic lipids on membrane surface via the polybasic motif (Figure 1 and Figure 2) and Ca^2+^ induced re-orientation of the same C2 domain away from the ring geometry disrupts the ring oligomers (Figure 4). […] The increased surface concentration of the E-Syt1 in the presence of Ca^2+^ could explain the improvement in the number of E-Syt1 rings observed under these conditions (Figure 1—figure supplement 2).”

5) In Figure 4 and Figure 4—figure supplement 1, the ring structures were disrupted after calcium addition. However, previous works show that there is a synergy between Syt1 (K326, K327)- PIP2 interaction and Syt1 (calcium-binding loops)-PS interaction (van den Bogaart et al. 2012, JBC). Moreover, the interaction between polybasic stretch (K326, K327) of Syt1 C2B and PIP2 has been reported not to be disrupted during calcium addition in the presence of t-SNARE/cis-SNARE complex (Wang et al. 2016, eLife). It is sometimes possible that the negative stain process affects the sample. Therefore, it is essential to reproduce the results with cryo-EM for samples treated by calcium in a short time before vitrification, to confirm whether Syt1 proteins stand around the disappearing ring structures similar to that observed with Syt2 in negative stain (Figure 1—figure supplement 2).

We had previously confirmed using cryo-EM (Wang et al.2014 Figure S3) that the negative stain process does not affect the Syt1 ring formation or its Ca^2+^ sensitivity (Wanget al. 2014). Further, we have shown that the disruption of the Syt1 ring was reversed when Ca^2+^ was chelated with EDTA (Wang et al.2014 Figure S8) confirming that the Syt1 proteins are still bound to the membrane following Ca^2+^ treatment.

*6) In the Discussion, the authors mentioned that "However, the molecular details of this interaction remains unclear, specifically due to the promiscuity and the low affinity of the Syt1/t-SNARE interaction". In recent work (Wang et al. 2016, eLife), it has been shown that the disassociation constant (Kd) between Syt1 and SNARE complex is about 1 μM in the presence of acidic membrane, compared to a large value obtained in the absence of membrane. The authors should cite this when discuss the functional relevant issues between Syt1 and t-SNARE.*

7) Zhou et al. (Nature, 2015), showed that one of the interfaces between the synaptogtamin C2B domain and the SNARE complex is essential for fast neurotransmitter release and forms prior to calcium injection. Are the proposed models of Syt1 oligomers compatible with this finding? The authors are kindly encouraged to provide a balanced discussion of the recent findings by Wang et al., 2016 and Zhou et al. 2015 and discuss their models in the context of these recent findings.

As recommended by the reviewers , we have now amended the discussion on Syt1 and t-SNARE interaction to read as follows: “Besides membranes, Syt1 also binds to t-SNAREs and this interaction is functionally relevant for fast neurotransmitter release (de Witet al. 2009, Mohrmannet al. 2013, Zhouet al. 2015, Wanget al. 2016). […] We note that in our Syt1 ring oligomer model, this binding interface on the C2B (marked as R398/R399 in Figure 5—figure supplement 1) is accessible and free to interact with the SNAREs. However, the occupancy and positioning of the SNARE complexes on the Syt1 ring oligomer is not known and as such, is the focus of our ongoing research.”

8) The ring-oligomers formed by the E-Syts were not disrupted by calcium what are the implications?

Reversible oligomerization might be critical to C2 domain protein functioning in calcium-triggered exocytosis, but not relevant for E-Syts function in maintaining the ER-PM appositions. We suppose that the ring-like oligomerization might stabilize the ER-PM contact sites and aid in the lipid transfer function of E-Syts. As such, Ca^2+^ insensitivity might allow the E-Syts to either maintain (E-Syt2) or even enhance (E-Syt1) its function upon Ca^2+^ influx. This is consistent with the functional role of E-Syts. However, more research is needed to address this in detail.

9) In the legend for Figure 1—figure supplement 2, the authors mention that: "In fact, calcium improved the number of rings observed with E-Syt1 consistent with the fact effective membrane interaction of E-Syt1 requires calcium (Ref)." It is unclear as to what is meant here, since calcium stimulates the binding of Syts and Doc2 to membranes, as well.

Addressed above in Point 4. The corresponding section in the figure legend has been revised to read as follows: “Ca^2+^ addition had divergent effect on the E-Syt isoforms, while the E-Syt2 was largely un-affected, the E-Syt1 rings were stabilized (discussed below).”

*10) It is unclear how many syt molecules are required to form the ring-like oligomers: 15? More than 15? What is the variation? What is the relevance to the copy number of syt1 per synaptic vesicle? The number of syt1 molecules per vesicle should be discussed, and (Takamori et al. 2006) cited. If it the required number is more than 15, it is highly unlikely that such rings form* in vivo *considering the protein number in synaptic vesicles. Moreover, given the physical constraints in the system (length of the linker that connects the C2-domains to the SV membrane etc.), more through and balanced discussion over whether syt1 can physically form these structures is required.*

Based on the helical indexing of the Syt1^C2AB^ decorated tubes (Wang et al. Figure 3), we estimate that the Syt1 ring with average outer diameter of ~ 30 nm corresponds to ~ 17 copies of Syt1 molecules. NOTE: Syt1^CD^ does not assemble in tubular structures and thus, we are unable to perform similar reconstruction with Syt1^CD^ ring oligomers. However, the rings assembled with Syt1^CD^ and Syt1^C2AB^ are remarkably similar and thus, the Syt1^C2AB^ tubes provide a very close approximation for the quantitative analysis. A synaptic vesicle contains ~16-22 copies of Syt1 molecules (Takamoriet al. 2006, Wilhelmet al. 2014), enough to form a ring oligomer of ~27 -37 nm in diameter, consistent with our EM analysis. Therefore, we believe that the rings of average diameters observed in our EM analysis can form at the SV-PM interface in vivo. We have now included the quantitative analysis in the Results (subsection “Complete Cytoplasmic Domain of Syt1 forms Rings under Physiologically Relevant Conditions”) and in the Discussion section (first paragraph).

*11) Given the physical constraints* in vivo *(e.g. membrane tethers, the diameter of synaptic vesicles etc.), can these rings form in vivo? And it might be wise to reexamine how the SNARE fit in, as they would seem to fit INSIDE these very large rings?*

Our data indicates that the Syt1 rings can assemble under physiological ionic strength and membrane composition. Given the length/flexibility of the juxtamembrane linker region, it should not affect the C2 ring formation, even when anchored in the synaptic vesicle. In fact, mathematical modeling faithfully reproduces the Syt1 oligomerization at the vesicle-plasma membrane interface under physiological conditions considering the vesicle dimension and the physical constraints, in support of the ‘washer’ model. A manuscript describing these findings is currently under preparation (Zhu, O’Shaughnessy & Rothman).

In Figure 5, the SNARE protein are shown outside the Syt1 rings (with no interaction) for illustrative purpose only to highlight that the height of ring would prevent the complete zippering of SNAREs. The occupancy and the orientation of the SNAREs on the Syt1 rings is not known and as such, focus of our current work. We have now included the following statement “The positioning and occupancy of SNAREs on Syt1 ring is not known and are shown for illustrative purposes only” (Figure 5 legend) to clarify the issue.

We favor the SNARE assembling outside of the Syt1 rings in the ‘washer model’ for the following reasons: 1) The inner dimensions of the Syt1 ring (~20 nm) could fit a small number of SNAREs, but not the multiple SNARE complexes (high copy number) that are needed for ultra-fast, evoked neurotransmitter release (Acuna et al. 2014). 2) Complexin is required for normal Ca^2+^ triggered release and biochemical/structural studies (Krishnakumaret al. 2011, Kummelet al. 2011) suggests that Complexin functions by organizing the pre-fusion SNARE complexes into a zig-zag array via *trans* interaction. It is not feasible to fit the inter-connected array of SNAREs inside the Syt1 rings. 3) Also, if the SNAREs were to zipper outside the Syt1 rings, then ring oligomers function as a ‘washer’ or spacer more effectively to impede the SNARE assembly and thus, synchronize fusion to Ca^2+^ influx.

12) A large conformational switch in syt1 was recently reported (e.g. Bai et al., 2016), and this report seems congruent with the model put forward in the current study, so this paper should be discussed and cited.

We thank the reviewer for the suggestion and we have now cited and discussed this report in the Discussion section (second paragraph) as follows: “This is congruent with the recent report (Bai et al. 2016), showing that the switch between the functional states (clamped vs. activated) of Syt1 involves large conformational change in the C2 domains.”

13) Complexin as a fusion clamp – the new review from Trimbuch & Rosenmund (2016) should be cited, as there is strong evidence complexin is not a clamp in murine neurons.

This reference has now been included and cited (Introduction, third paragraph).

14) The observation of E-Syt rings is very interesting. The lipid transfer activities of E-Syts remain intact when their C2 domains are replaced by those of Syt-1, consistent with the authors' model. However, the lipid-harboring synaptotagmin-like-mitochondrial-lipid binding protein (SMP) domains were not included in this study. Are these SMP domains compatible with the configuration of the ring if they all face inside?

The SMP domain are connected to the C2 domains via a flexible linker, that is believed to be long enough to allow them shuttle between the ER and PM for lipid transfer (Schauderet al. 2014, Reinisch and De Camilli 2015). So, the SMP domains can locate outside of the E-Syt ring i.e. positioning of the SMP domain is not limited by the size of the E-Syt rings. X-ray structure of E-Syt2 shows that the SMP domain self-dimerize, however it is not clear if the in vivo functional form is a dimer, as other TULIP domain could function as a monomer (Reinisch and De Camilli 2015). Nevertheless, the dimension of the SMP dimer (~9 nm) (Schauderet al. 2014) should fit within the inner diameter of the E-Syt rings (~20 nm). We require additional studies including SMP domain to understand how its influences the C2 domain organization, and how the oligomerization would affect its function.

15) Does PIP2 influence the ring-forming properties of other C2 domain molecules such as Doc2b and E-Syts?

The ring assembly of Syt1 is triggered by the binding of the polylysine motif of the C2B domain to the anionic lipid on the lipid surface. Under the right condition, the identity of the anionic lipid does not appear to be a critical factor as we observe Syt1 rings on monolayers containing PS alone or PIP2 alone. However, under physiologically relevant conditions, the interaction with PS will be limited and thus, PIP2 will be a critical factor. We believe that this will be true for other C2 domain molecules such as Doc2B and E-Syts. We limited our analysis with PIP2 containing monolayers to Syt1 due to the technical challenges involved in forming stable PIP2 containing monolayers on the EM grids.

16) The authors propose that the E-Syt rings prevent membrane fusion at ER-PM contact sites. This seems unlikely because membrane fusion is not a spontaneous process that needs to be "prevented". Instead, the ER-PM contact sites may simply lack the membrane fusion machinery. In support of this notion, lipid transfer also occurs at other membrane contact sites where no C2 domain molecules are found. These concepts need to be addressed by the authors.

We agree and so we have now re-worded this sentence to read: “In particular, E-Syts, which function to enable the ER and plasma membrane to come into intimate contact – close enough for lipids to be transferred. Our results suggest this might be achieved by bridging two membranes with an intervening structure, most probably based on ring oligomers”

17) Generally, the results with E-Syt are among the most interesting new results in this paper, but yet the discussion almost exclusively focuses on Syt1 (which is more controversial in view of the copy number in synaptic vesicles and other concerns raised in the above points). Thus, the authors are kindly asked to expand the discussion of the E-Syt results and their biological implications.

We share the reviewer’s enthusiasm on the commonality of the ring oligomers of C2 domain protein, in particular with distantly related proteins like E-Syts and intrigued by its implication on the organization and function of E-Syts. As described in Point 14, the ring oligomers are compatible with SMP domain, a key functional element of the E-Syts, which are believed to self-dimerize (Schauderet al. 2014). However, more research, particularly with constructs which include the SMP domain, is required to address this adequately. We have thus limited our discussion on E-Syts, but rather focused on Syt1 for which we have ample data to support a functionally relevant role. We are confident that we addressed many of the concerns raised by the reviewers, particularly on the synaptic vesicle copy number, in the revised manuscript. Nevertheless, we have now expanded the discussion on the Ca^2+^ sensitivity of the E-Syts and refer to possible role of ring oligomers at ER-PM junctions in the Discussion section (fourth paragraph).

Optional revisions:

*18) As Honigmann et al. 2013, NSMB and van den Bogaart et al. 2011, Nature reported, PIP2 displays a roughly 73 nm-sized microdomains in company with syntaxin-1a. In this work, however, there is no syntaxin reconstituted on monolayer. It is conceivable that Syt1 binds preferentially to PIP2 rather than other acidic lipids (e.g. PS) in the absence of Ca^2+^. How can the ~30 nm-sized ring structure be stabilized? What is the morphology of the ring structure in the presence of syntaxin and PIP2 without calcium?*

*19) Are the ring-like structures of Syt-1 affected by complexin and/or t-SNAREs (e.g., anchored to monolayers through Nickel-His6 interactions)?*

Under physiological conditions, the Syt1-PIP2 interaction mediated via the polybasic motif positions the C2B domains on the membrane and facilitates the circular oligomerization of the Syt1 molecule. Modeling studies (Figure 5—figure supplement 1) shows that this involves spatial organization of the C2B domain with the Ca^2+-^binding loops locating at the interface. We currently do not have high resolution structure to identify the molecular contact that stabilize the ring oligomers. Recent report (Wanget al. 2016) shows that Syt1 can bind PIP2 and t-SNARE simultaneously and our reconstruction shows that the primary t-SNARE binding site (marked as R398/R399 and shown in green in Figure 5—figure supplement 1) as identified in Zhou et al.Nature 2015 is available for interaction. So, we concur that the binding SNARE complex (via t-SNAREs) or Complexin should not interfere with the ring assembly. However, additional research is required to be certain.